# A moth odorant receptor highly expressed in the ovipositor is involved in detecting host-plant volatiles

**Rui-Ting Li[1,2], Ling-Qiao Huang[1], Jun-Feng Dong[3], Chen-Zhu Wang[1,2]***

[1]State Key Laboratory of Integrated Management of Pest Insects and Rodents, Institute of Zoology, Chinese Academy of Sciences, Beijing, China; [2]CAS Center for Excellence in Biotic Interactions, University of Chinese Academy of Sciences, Beijing, China; [3]Forestry College, Henan University of Science and Technology, Luoyang, China

**Abstract** Antennae are often considered to be the nostrils of insects. Here, we sequenced the transcriptome of the pheromone gland-ovipositor complex of *Helicoverpa assulta* and discovered that an odorant receptor (OR) gene, *HassOR31*, had much higher expression in the ovipositor than in antennae or other tissues. To determine whether the ovipositor was involved in odorant detection, we co-expressed *HassOR31* and its co-receptor, *HassORco*, in a *Xenopus* oocyte model system, and demonstrated that the OR was responsive to 12 plant odorants, especially Z-3-hexenyl butyrate. These odorants elicited electrophysiological responses of some sensilla in the ovipositor, and *HassOR31* and *HassORco* were co-expressed within ovipositor sensilla. Two oviposition preference experiments showed that female moths lacking antennae still preferentially selected oviposition sites containing plant volatiles. We suggest that the expression of *HassOR31* in the ovipositor of *H. assulta* helps females to determine precise egg-laying sites in host plants.

**\*For correspondence:**
czwang@ioz.ac.cn

**Competing interests:** The authors declare that no competing interests exist.

## Introduction

The most important functions of adult insects are to find optimal mates and suitable habitats for survival and success of their offspring. The insect olfactory system plays a key role in these processes, and antennae are often considered to be the nostrils of insects. However, some species also use other cephalic organs, such as maxillary palps and proboscis, to detect volatile compounds (*Haverkamp et al., 2016*; *Di et al., 2017*; *Lu et al., 2007*). Olfactory sensilla distributed on these organs are multiporous hair-like structures innervated by the dendrites of olfactory sensory neurons (OSNs) (*Benton and Dahanukar, 2011*; *Stocker, 1994*). Odorant receptors (ORs) are atypical, 7-transmembrane domain proteins, which are located on the dendritic membrane of OSNs, selectively bind to volatile ligands in the environment and are the primary determinants of the detection spectrum of OSNs (*Dobritsa et al., 2003*; *Goldman et al., 2005*; *Störtkuhl and Kettler, 2001*; *Wicher et al., 2008*). These ligand-binding ORs are considered to form a heteromultimeric complexes with a co-receptor (ORco) and to function as non-selective cation channel (*Butterwick et al., 2018*; *Larsson et al., 2004*; *Neuhaus et al., 2005*; *Sato et al., 2008*; *Vosshall et al., 1999*). Some ionotropic receptors (IRs) and gustatory receptors (GRs) are also recognized as odorant-detecting receptors (*Benton et al., 2009*).

Several recent studies have challenged the hypothesis that only cephalic organs are involved in the detection of volatile compounds in insects. Olfactory receptor genes are expressed not only in antennae and maxilla, but also in ovipositors of a variety of moth species, including *Heliothis virescens* (*Widmayer et al., 2009*), *Sesamia nonagrioides* (*Glaser et al., 2013*), *Chilo suppressalis*

**eLife digest** When most insects reproduce they lay eggs that hatch into juveniles known as larvae. To provide good sources of food for the larvae, the adult insects have to carefully select where to lay the eggs. Host plants produce specific sets of chemicals known as odorants that the adult insects are able to smell using proteins called odorant receptors.

It is generally thought that odorant receptors in the antennae on the head are responsible for guiding adult insects to good egg-laying sites. However, recent studies have reported that odorant receptors are also present in the egg-laying organs of several different species of moth. It remains unclear what role these odorant receptors may play in egg-laying.

The oriental tobacco budworm (*Helicoverpa assulta*) is considered a serious pest in agriculture. The adult moths lay their eggs on a narrow range of plants in the nightshade family including tobacco and hot pepper. Li et al. have now investigated the odorant receptors of *H. assulta* and found that one gene for an odorant receptor called *HassOR31* was expressed much more in the egg-laying organs of the moths than in the antennae. Further experiments showed that this receptor was tuned to respond to 12 odorants that also stimulated responses in the egg-laying organ of *H. assulta*. Together these findings suggest that this odorant receptor in the egg-laying organ helps the moths find suitable host plants to lay their eggs on.

The work of Li et al. may help us understand how *H. assulta* evolved to lay its eggs on specific members of the nightshade family and lead to new methods of controlling this pest. An insect's sense of smell guides many other behaviors including finding food, mates and avoiding enemies. Therefore, these findings may inspire researchers to investigate whether odorant receptors in the antennae or other organs guide these behaviors.

(*Xia et al., 2015*), and *Manduca sexta* (*Klinner et al., 2016*). However, the function of olfactory receptor genes expressed in the ovipositor remains unknown.

Moth ovipositors are always connected to pheromone glands, and several lines of evidence from different moth species suggest that functional olfactory receptors may be present in the sensilla located on moth ovipositors. For example, a pheromone receptor is expressed in the ovipositor of *H. virescens*, suggesting a possible role of the ovipositor in feedback regulation of biosynthesis and emission of the female sex pheromone from the sex pheromone gland (*Widmayer et al., 2009*). Two ORs are expressed in the pheromone gland and ovipositor of the grassland moth, *C. suppressalis*, but no ORco was detected there (*Xia et al., 2015*). Sensilla with a multiporous surface are observed in *Monopis crocicapitella* and *Homoeosoma nebulella* (*Faucheux, 1991*; *Faucheux, 1988*). Finally, a subset of sensilla located on the ovipositor of *M. sexta* exhibit electrophysiological responses to a large array of volatile organic compounds, and expression of ORco and the ionotropic co-receptors IR8a and IR25a was detected in the ovipositor (*Klinner et al., 2016*). However, the function of ORs expressed in the ovipositor of moths and whether these ORs are co-expressed with ORco remains unclear.

The Oriental tobacco budworm, *Helicoverpa assulta*, is a serious crop pest with a narrow host plant range which includes only Solanaceae such as tobacco, hot pepper, and several *Physalis* species (*Wang et al., 2004*). Previous antennae transcriptome studies showed that more than 60 ORs were expressed in the antennae of *H. assulta* (*Xu et al., 2015*; *Zhang et al., 2015*). The function of the pheromone receptors and several ORs responding to plant volatiles were characterized (*Cao et al., 2016*; *Wu et al., 2019*; *Yang et al., 2017*). To determine whether any ORs were expressed in the ovipositor, we sequenced the transcriptome of the pheromone gland-ovipositor complex and found one OR with a very high expression level. Next, we functionally analyzed the response spectra of this OR to a wide range of host plant-related odorants using the *Xenopus* oocyte expression system and two-electrode voltage-clamp recording. In situ hybridization, scanning electron microscopy and electrophysiology studies indicated that this OR is expressed in some multiporous sensilla on the moth ovipositor. Together with the results of oviposition experiments, we suggest that the OR expressed in the ovipositor helps *H. assulta* females to find precise egg-laying sites on their host plants.

# Results

## Transcriptome sequencing and identification of chemosensory receptors in pheromone gland - ovipositor of *Helicoverpa assulta*

We conducted next-generation transcriptome sequencing analyses using a cDNA library constructed from pheromone gland-ovipositors (PG-OVs) of female *H. assulta* using the Illumina HiSeq 2000 platform. We downloaded amino acid sequences of odorant receptors, gustatory receptors, antennal ionotropic receptors, ionotropic glutamate receptors, odorant binding proteins, chemosensory proteins, general odorant binding proteins, pheromone binding proteins and sensory neuron membrane proteins of *H. assulta* and *H. armigera* from NCBI to construct a local database (*Liu et al., 2014*; *Liu et al., 2018*; *Tillman et al., 1999*; *Xu et al., 2015*; *Zhang et al., 2015*), and then performed a BlastX search against this database to identify the cDNA sequences of chemosensory receptors from the transcriptome. The transcripts of 22 *OR*s, 6 *GR*s, 13 *IR*s, and 9 *iGluR*s were detected (*Figure 1A*). *Figure 1—figure supplement 1* and *Supplementary file 1* shows the identified putative chemosensory related genes expressed in PG-OVs and their values of TPM (Transcripts Per Kilobase of exon per Million mapped reads). *HassOR31* and *HassiGluR7* had approximately the same TPM value which was the highest among those of all the chemosensory receptor genes. Most crucially, *HassORco* was also detected but its TPM value was much lower than that of *HassOR31*.

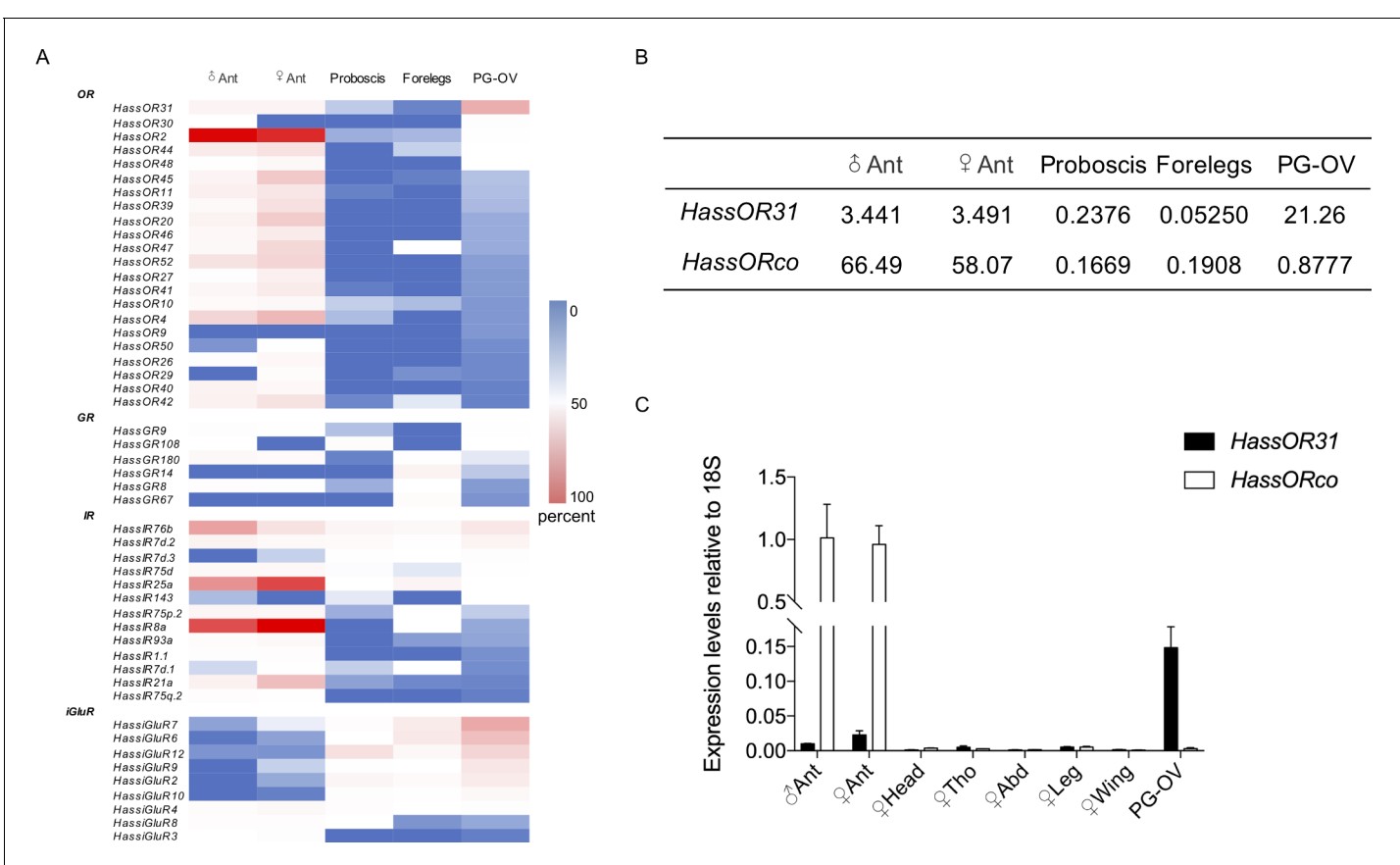

**Figure 1.** Gene expression of *HassOR31* in different tissues of *H. assulta*. (**A**) Tissue expression profiles of putative chemosensory receptor genes identified in the pheromone gland-ovipositor complex (PG-OV) of *H. assulta*. (**B**) TPM values of *HassOR31* and *HassORco* in different tissues. Ant, antennae; Proboscis, mixed female and male proboscis; Forelegs, mixed female and male forelegs. (**C**) qRT-PCR results of *HassOR31* and *HassORco* in different tissues relative to housekeeping gene *18S*. Tho, thorax; Abd, abdomen.
The online version of this article includes the following source data and figure supplement(s) for figure 1:

**Source data 1.** Source data for *Figure 1A and C*.
**Figure supplement 1.** Tissue expression profiles of some putative chemosensory related genes identified in the pheromone gland-ovipositor complex of *H. assulta*.

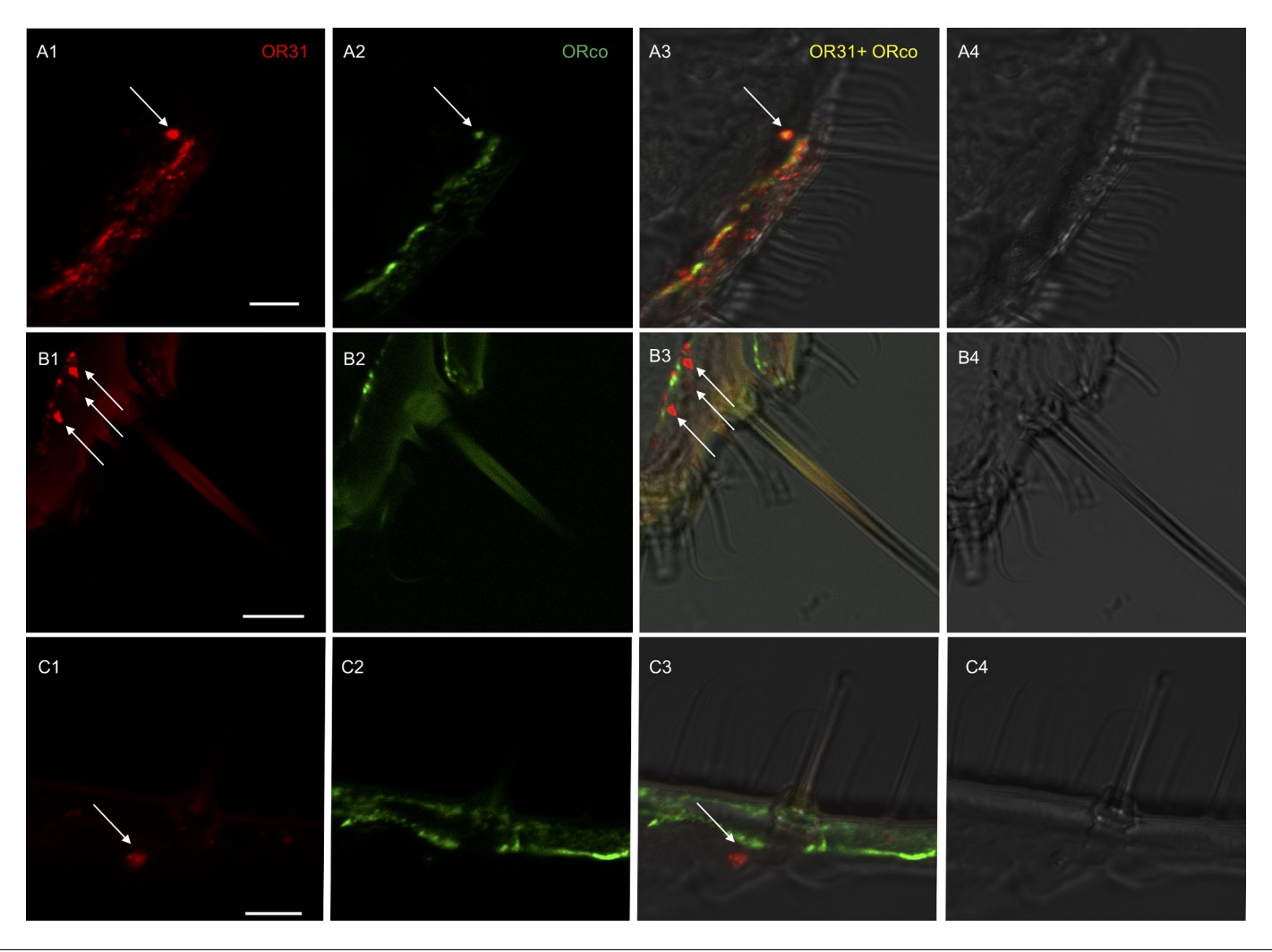

**Figure 2.** Localization of *HassOR31* and *HassORco* expression in cells of *H. assulta* ovipositors. Double-FISH with female ovipositors using combinations of labeled *OR* probes and visualization of cells bearing distinct *HassOR31* transcripts by red (DIG) (**A1, B1, C1**) and *HassORco* transcripts by green (biotin) (**A2, B2, C2**) fluorescence, respectively. Co-labelling of cells by both OR probes appear as yellow/orange color in the overlay of the red and green fluorescence channels (**A3, B3, C3**). Bright-field images are presented as references (**A4, B4, C4**). Arrows indicate the cell location. Scale bars: 10 μm. (**A1-3**) The *HassOR31* and *HassORco* probes label the same cell. (**B1-3, C1-3**) Only the *HassOR31* probe is detected. Scale bars: 10 μm. The online version of this article includes the following figure supplement(s) for figure 2:

**Figure supplement 1.** More examples of two-color in situ hybridization visualizing the combinations of *HassOR31* (red) and *HassORco* (green) in the ovipositor of *H. assulta*.

## Tissue expression pattern of *HassOR31*

Therefore, we further analyzed the tissue expression profiles of *HassOR31* and *HassORco* by qRT-PCR. The qRT-PCR results indicated that HassOR31 had the highest expression level in PG-OV, which was 7 times higher than that in female antennae and 15 times higher than that in male antennae. *HassORco* was also expressed in PG-OV, but the expression level was much lower than that in the antennae of both sexes (*Figure 1C*). In PG-OV, when Ct value of *Hass18S* was 20.41, the Ct values of *HassOR31* and *HassORco* were 23.18, 28.78, respectively. These results were in line with the RNA-seq data (*Figure 1B*).

## Localization of *HassOR31* and *HassORco* in the ovipositor

To further determine which cells were expressing *HassOR31* and *HassORco* in the ovipositor, two-color double in situ hybridization experiments were performed with Dig-labeled *HassOR31* and Bio-labeled *HassORco*. In some cases, *HassOR31* and *HassORco* were co-expressed within the cells beneath certain sensilla with a large size (*Figure 2A*, *Figure 2—figure supplement 1*, *Video 1*), while in most cases *HassOR31* was expressed alone in the cells beneath small hairs, but a few in the cells of sensilla with a large size (*Figure 2B and C*, *Figure 2—figure supplement 1*, *Video 1*), which is consistent with the high expression of *HassOR31* and the low expression of *HassORco* in PG-OV of *H. assulta*.

## Functional analysis of HassOR31 by *Xenopus laevis* oocytes

A *Xenopus* oocyte expression system with two-electrode voltage-clamp recording was used to characterize the function of *HassOR31*. A panel of 51 chemicals with behavioral or electrophysiological activities to *Helicoverpa* species as listed in *Supplementary file 3* were used to screen the ligands of *HassOR31* (*Di et al., 2017*). They were classified into four categories, green leaf volatiles (GLVs), terpenoids, aromatics, and aliphatics. The oocytes containing co-expressed HassOR31/HassORco were tuned to 12 odorants, including Z-3-hexenyl-butyrate, myrcene, citral, and Z-3-hexenyl acetate, with Z-3-hexenyl-butyrate as the most effective ligand (*Figure 3A,B*). The oocytes in which only *HassOR31* was expressed had no

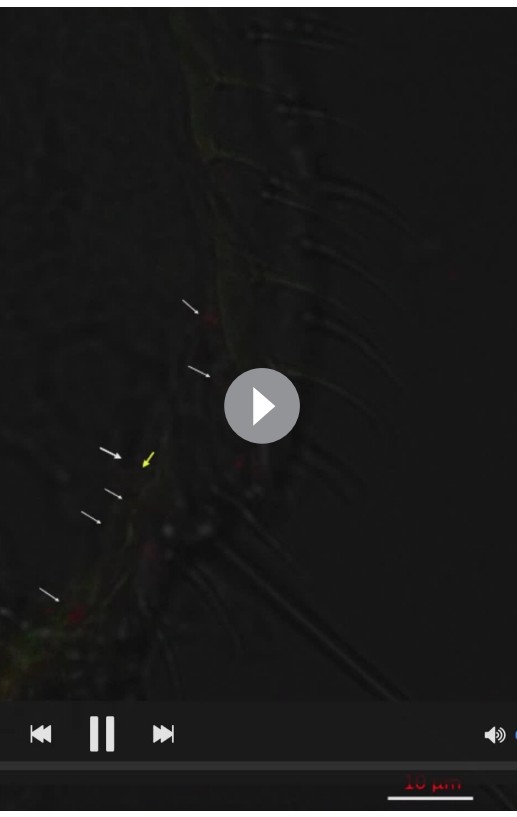

**Video 1.** Z-stack video of two-color in situ hybridization visualizing the combinations of *HassOR31* (red) and *HassORco* (green) in the ovipositor of *H. assulta*. White arrows indicate the cells only expressing *HassOR31*, and yellow arrow the cell co-expressing *HassOR31* and *HassORco*.

https://elifesciences.org/articles/53706#video1

positive responses to these odorants. Since *HassiGluR7* had a high expression level comparable to *HassOR31* in the pheromone gland - ovipositor, we also co-expressed *HassOR31* and *HassiGluR7* in the oocytes, but no positive responses were detected (*Figure 3A*).

## Scanning electron micrographs (SEM) of the ovipositor of *H. assulta* and its associated sensilla

To further determine the putative chemosensory sensilla which may bury HassOR31/HassORco on the ovipositor of *H. assulta*, we examined the surface of the anal papillae using scanning electron micrography (*Figure 4*), and found four types of sensilla (*Figure 4A,B*). The first type is long sensilla with smooth non-porous surfaces, sharp tips and raised sockets, which may have a potential function in mechanosensation (*Figure 4C,D*). The second type is shorter sensilla with non-porous surfaces and sunken sockets, which may also have a mechanical function (*Figure 4E,F*). The third type sensilla is of similar length with the second type. It is morphologically like trichoid sensilla in the antennae, with pores on the surface and a single pore at the tip (*Figure 4G,H*), and therefore may have a function in olfaction and/or taste. They are mainly distributed on the middle part of the ovipositor. Finally, the fourth type sensilla are morphologically similar to basiconic sensilla in the antennae, with pores on the surface and a large pore at the tip and may also have a function in olfaction and/or taste (*Figure 4I,J*). They are located near the ovipore and five to seven of them are distributed on each papilla. Moreover, the papillae are covered with short, poreless microtrichia.

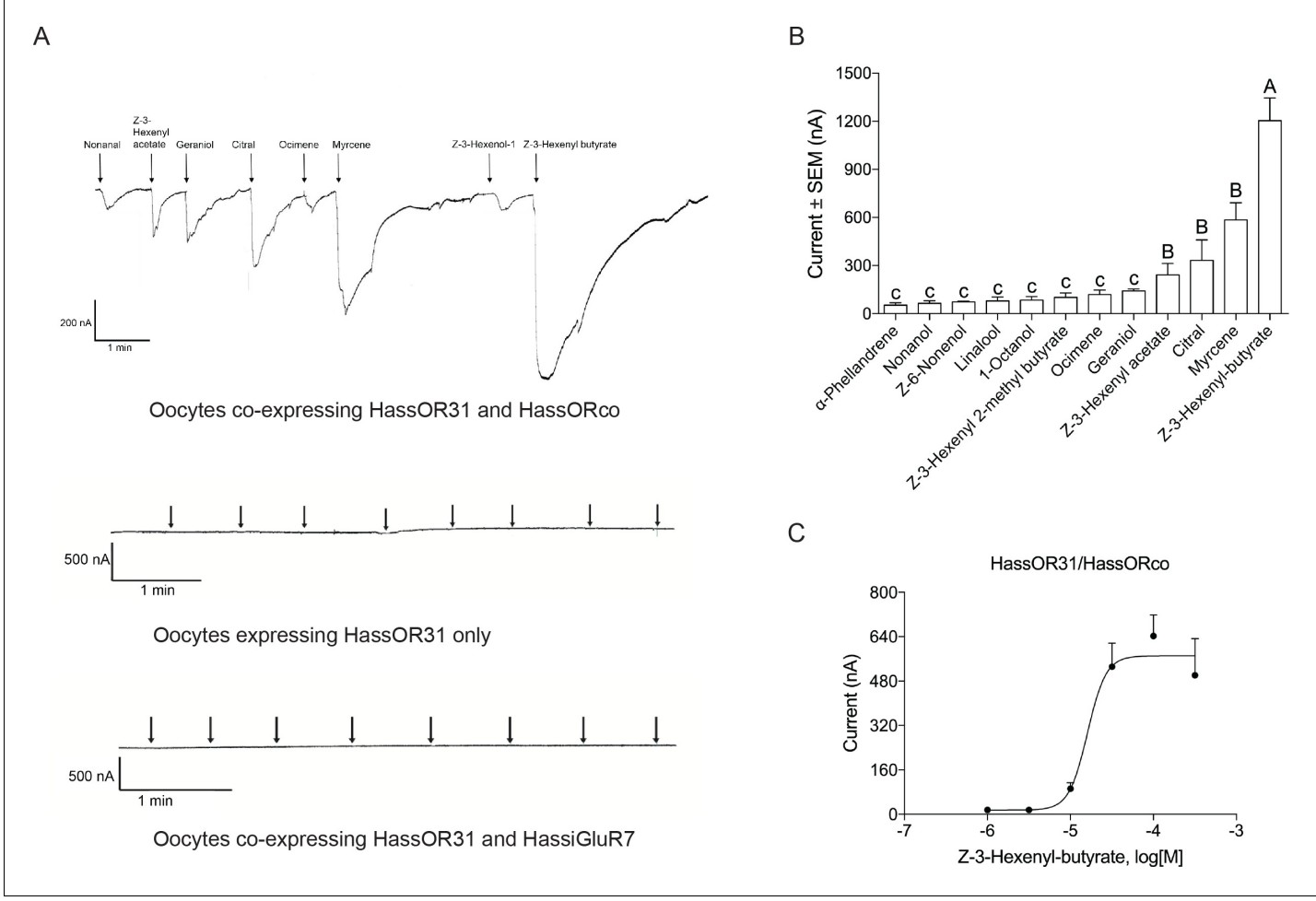

**Figure 3.** Functional analyses of chemosensory receptors in *Xenopus* oocytes. (**A**) Inward current responses of *Xenopus* oocytes with expressed HassOR31/HassORco, HassOR31 alone, or HassOR31/HassiGluR7 in response to odorants ($10^{-4}$ M solution). Odorants were applied for 8 s at times indicated by arrowheads; (**B**) Odorant-response spectra of *Xenopus* oocytes with expressed HassOR31/HassORco. Responses were measured as induced inward currents, expressed in nA. Error bars show standard error of the mean (n = 3–5), columns with different letters are significantly different at p<0.05 (One-way ANOVA followed by post hoc analysis with Turkey test); (**C**) Dose responses of *Xenopus* oocytes with expressed HassOR31/HassORco to the most effective ligand Z-3-hexenyl butyrate (n = 4), The EC$_{50}$ value for Z-3-hexenyl butyrate was $1.606 \times 10^{-5}$ M.

The online version of this article includes the following source data for figure 3:

**Source data 1.** Source data for *Figure 3B and C*.

## Single sensillum recordings (SSR) of putative chemosensory sensilla

We determined whether these identified putative chemosensory sensilla served an olfactory function using single sensillum recordings. Seventeen odorants, including the major ligands of HassOR31/HassORco and sex pheromone components of *H. assulta*, were used as stimuli (*Supplementary file 3*). Most tested sensilla showed baseline spiking activity, several sensilla responded to the tested odorants (*Figure 5A*). Z-3-Hexenyl-butyrate, the most effective ligand of HassOR31/HassORco, gave a dose-dependent response curve (*Figure 5B,C*).

## Oviposition preference of *H. assulta* female

Based on the above results, we hypothesize that the ovipositor may play a role in oviposition site detection of *H. assulta* female. To verify this, we designed two oviposition choice tests (*Figure 6—figure supplement 1*). Intact, mated females preferred to lay eggs on the areas of gauze exposed to host plant volatiles (n = 7; p=0.0008) (*Figure 6A*). The closer to the odor source, the higher the density of eggs on the gauze (*Figure 6—figure supplement 1*). Mated females whose antennae had

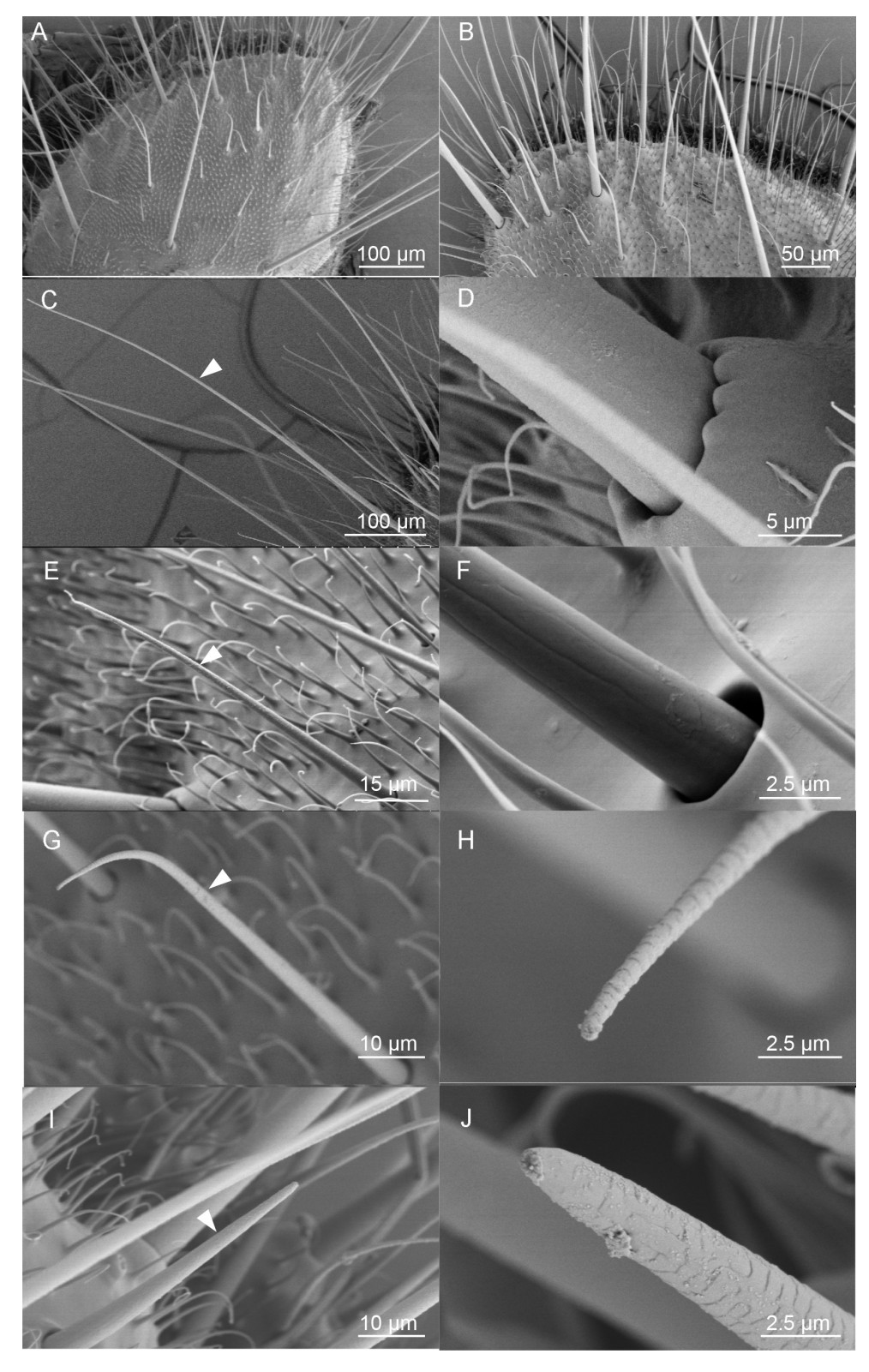

**Figure 4.** Scanning electron micrographs (SEM) of the ovipositor of *H. assulta* and its associated sensilla. (**A**) An overview of the extended tip of the ovipositor of *H. assulta*, which contains two anal papillae surrounding the ovipore. (**B**) Details of the papilla close to the ovipore. There are four types of sensilla distributed on each papilla: Type I (**C**), putative mechanical sensilla widely distributed on the papilla, with raised pocket-like bases and a *Figure 4 continued on next page*

*Figure 4 continued*

smooth surface (D); Type II (E), putative mechanical sensilla with sockets, showing flexible areas and non-porous surfaces (F); Type III (G), putative olfactory/taste sensilla shorter than Type I sensilla and morphologically similar to the trichoid on antennae, with pores on both the surface and the tip (H); Type IV (I), putative olfactory/taste sensilla located near the ovipore and morphologically similar to the basiconic sensilla, with pores on the surface and a large terminal pore (J).

been removed showed a reduced, but still significant preference for oviposition on volatile-treated areas (n = 7; p=0.0088) (*Figure 6A*, *Figure 6B*). Intact (n = 5; p=0.0053) and antennectomized (n = 5; p=0.0149) females both preferred to lay eggs on Z-3-hexenyl butyrate treated fake leaves (*Figure 6C*), while they showed no significant difference on the oviposition preference index (*Figure 6D*).

## Discussion

We successfully identified a functional OR expressed highly in PG-OV of *H. assulta* for the first time. The expression level of *HassOR31* in PG-OV was about five times higher than that in antennae of both females and males. *HassOR31* and *HassORco* were co-expressed beneath certain sensilla in the ovipositor. *Xenopus* oocytes containing expressed HassOR31/HassORco responded to twelve GLV and terpenoid compounds, which also elicited electrophysiological responses from some ovipositor sensilla. Finally, behavioral tests showed that besides the antennae, the ovipositor could be also involved in detecting plant volatiles and played a role in oviposition site selection of *H. assulta*.

### The function of HassOR31 expressed in the ovipositor

HassOR31 was relatively widely tuned to a spectrum of plant volatiles. The twelve GLV and terpenoid compounds are widely present in leaves, flowers, and fruits of plants, and are strongly attractive to herbivorous insects (*Gregg et al., 2010*). The ortholog of HassOR31, HarmOR31, has a similar

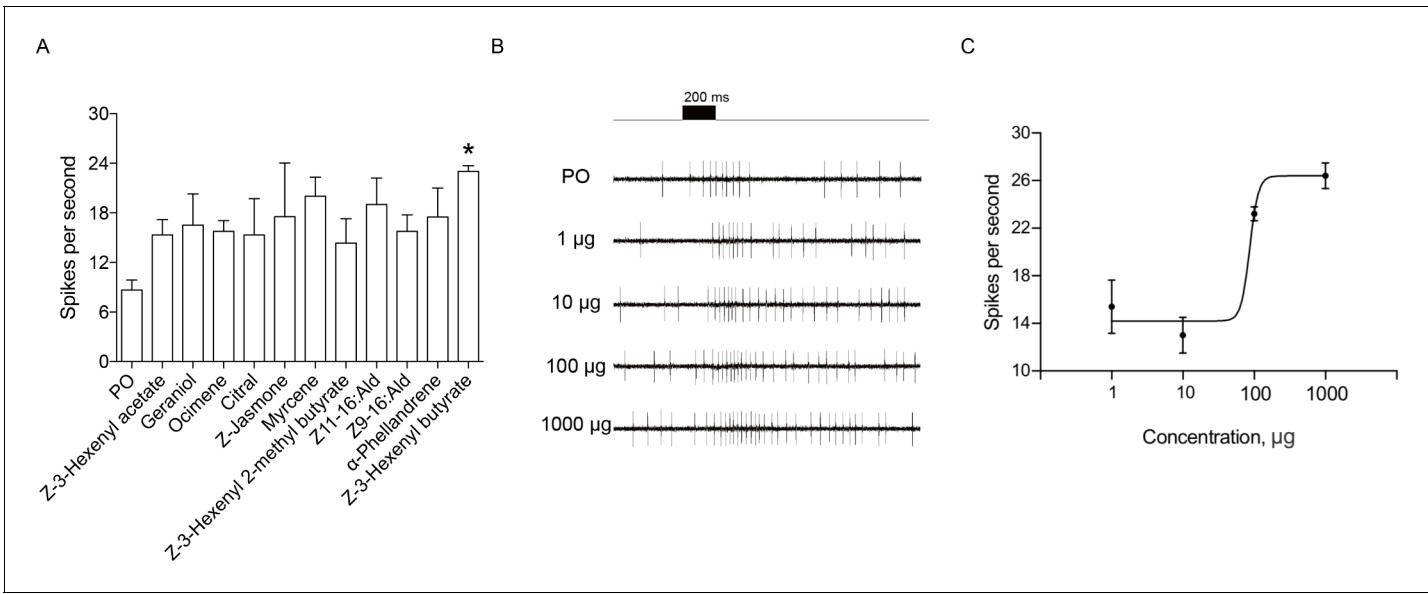

**Figure 5.** Single sensillum recordings (SSR) of putative chemosensory sensilla in an *H. assulta* ovipositor. (**A**) The firing rate of a putative chemosensory sensilla to several plant volatile and sex pheromone components. Chemicals with * induced significantly electrophysiological responses at p<0.05 relative to PO (paraffin oil) (One-way ANOVA followed by post hoc analysis with Turkey test, n = 3–4; Z-3-Hexenyl butyrate, p=0.0224). (**B**) The exemplary recordings of electrophysiological activities in a putative chemosensory sensilla to different load doses (1 µg, 10 µg, 100 µg and 1000 µg) of Z-3-hexenyl butyrate. (**C**) SSR dose responses of putative chemosensory sensilla to Z-3-hexenyl butyrate (n = 5), EC$_{50}$ = 86.50 µg.
The online version of this article includes the following source data for figure 5:

**Source data 1.** Source data for *Figure 5A and C*.

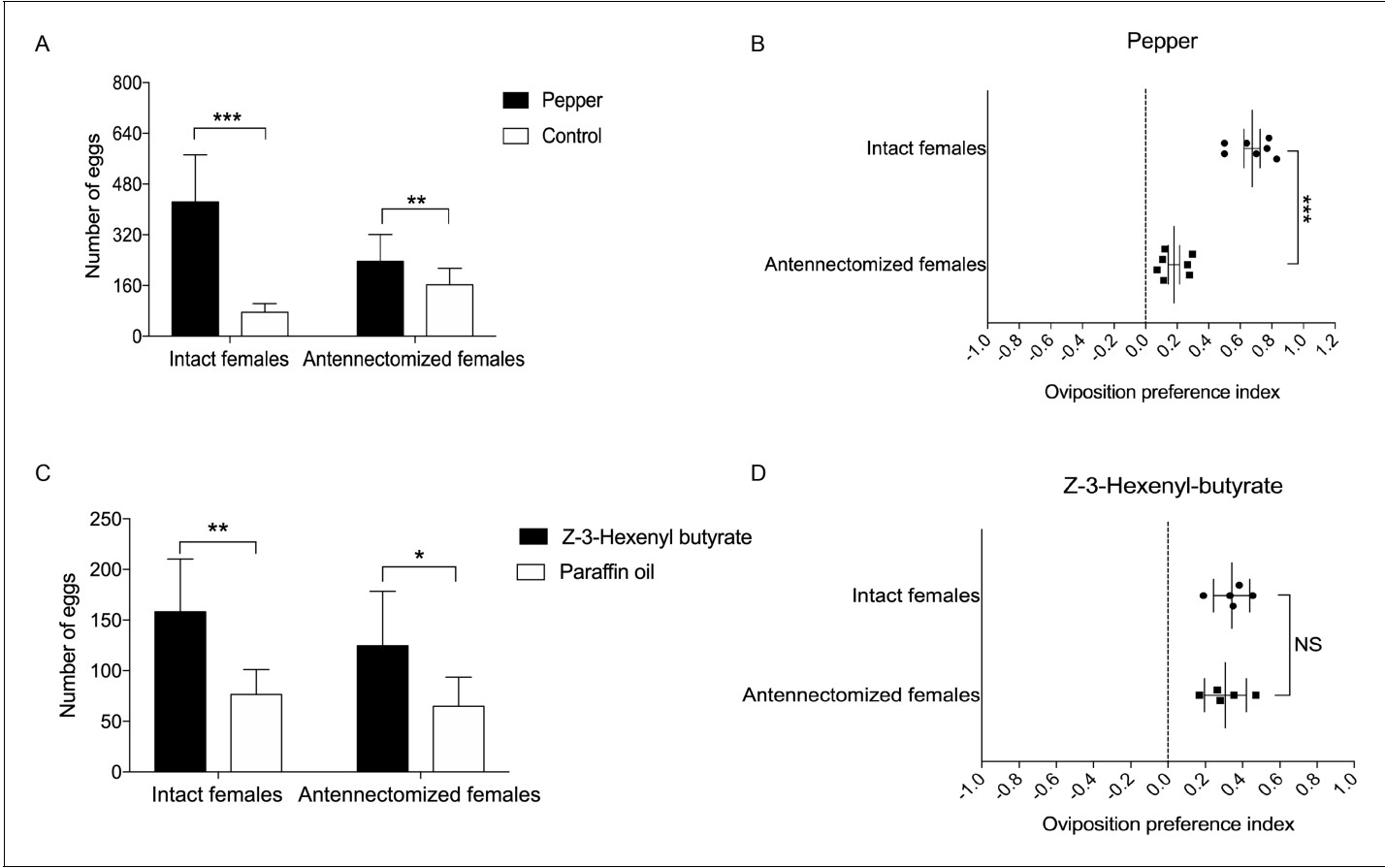

**Figure 6.** Oviposition preference of the antennae amputated and intact mated females of *H. assulta*. (A) Number of eggs on the hot pepper treated gauze and control gauze. Two and three asterisks indicate significant difference (p<0.01 and p<0.001, paired t test). (B) Oviposition preference index between the hot pepper treated gauze and control gauze. Three asterisks indicate significant difference (p<0.001, unpaired t test). (C) Number of eggs on the Z-3-hexenyl butyrate treated fake leaves and control fake leaves. One and two asterisks indicate significant difference (p<0.05 and p<0.01, paired t test). (D) The oviposition preference index between the Z-3-hexenyl butyrate treated fake leaves and control fake leaves. NS indicates no significant difference (p>0.05, unpaired t test). All experiments were carried out with five to seven biological replications, with 10–15 mated female moths used in each replicate.

The online version of this article includes the following source data and figure supplement(s) for figure 6:

**Source data 1.** Source data for *Figure 6A, B, C and D*.

**Figure supplement 1.** The set-up of oviposition choice tests and the spread of eggs laid by mated females of *H. assulta*.

wide spectrum in polyphagous *Helicoverpa armigera* (*Di et al., 2017*). Nevertheless, the most effective ligand of HassOR31 was Z-3-hexenyl butyrate while that of HarmOR31 was myrcene. The expression levels of the two orthologs in the ovipositors of the two species were also different: *HassOR31* was highly expressed while *HarmOR31* was poorly expressed (*Supplementary file 1*). This may have significance in the divergence of the host plant range between the two closely related species: *H. assulta* is a specialist on hot pepper, tobacco, and some *Physalis* plants in Solanaceae, while *H. armigera* is a typical generalist with a host plant range of over 300 species belonging to 68 plant families (*Pearce et al., 2017*; *Sun et al., 2012*; *Wang et al., 2004*).

As a specialist, *H. assulta* lays their eggs singly, preferably on or near the flowering or fruiting parts of these host plants (*Sun et al., 2012*; *Wu, 1990*). The headspace collections of tobacco flowers were analyzed by GC-MS and GC-EAD previously. Tobacco flowers release volatiles including E-β-ocimene, Z-3-hexenyl acetate, nonanal, Z-3-hexenyl 2-methyl butyrate, linalool, and Z-3-hexenyl butyrate, and blends of them are attractive to *H. assulta* females (*Sun et al., 2012*). Most odorants in the response spectrum of HassOR31/HassORco can be detected in headspace collections from tobacco.

As a general plant volatile and most effective ligand of HassOR31, Z-3-hexenyl butyrate is released by many plant fruits and has a green, fruity, and somewhat buttery aroma. It is also present in the headspace collections of tobacco flowers (**Sun et al., 2012**) and hot pepper fruits (**Forero et al., 2009**). Therefore, Z-3-hexenyl butyrate is very likely to be used by females of *H. assulta* as a signal for choosing oviposition sites rather than a signal for searching host plants. The behavioral assay with hot pepper volatiles (a complex cue) showed that antennectomized females still preferred to oviposit on sites treated with hot pepper volatiles, but their oviposition preference index was significantly decreased. Another behavioral assay with Z-3-hexenyl butyrate showed that antennectomized and normal females both prefer to lay eggs on the sites treated with the ligand for the receptor in the ovipositor but had no significant difference in oviposition preference index. These results suggest that detecting host volatile blends in general oviposition is governed by both antennae and ovipositor but detecting for Z-3-hexenyl butyrate seems to be mainly governed by the ovipositor. The expression levels of HassOR31 and other ORs expressed in the ovipositor and antennae may explain such behavioral responses. We speculate that a gravid female moth takes two steps to find an oviposition site: firstly, she smells the plant volatiles mainly by using antennae to search for a host plant, and secondly when she comes near to or land on the host plant, she integrates the information from olfactory sensilla as well as mechanical and contact chemosensory sensilla on the ovipositor to determine the precise oviposition sites on the host plants. Z-3-hexenyl butyrate and other ligand compounds of HassOR31 are expected to play a major role in the second step.

## Large difference in expression of *HassOR31* and *HassORco* in the ovipositor

It is believed that ORs cannot function in the absence of ORco (**Sato et al., 2008**). Recent Cryo-EM structure of the insect olfactory receptor ORco supports a model in which ORco and OR subunits assemble into a heterotetramer with a central shared ion-conduction pathway (**Butterwick et al., 2018**). One unresolved question from this study relates to the large difference in expression levels between *HassOR31* and *HassORco* in the ovipositor. We first assumed that *HassOR31* played a role alone without *HassORco*. To test this idea, we injected *HassOR31* cRNA alone into the *Xenopus* oocytes and found that oocytes expressing *HassOR31* alone had no response to the tested compounds. We then hypothesized that some co-receptor-like proteins might replace HassORco to function in cooperation with HassOR31. In addition to HassOR31, the ionotropic glutamate receptor, *HassiGluR7,* was also highly expressed in the ovipositors of *H. assulta* and had a similar tissue expression pattern to *HassOR31*. To test if these two receptors function together, we co-expressed *HassOR31* and *HassiGluR7* in *Xenopus* oocytes, but no response was detected.

In the electrophysiological experiments, we discovered only a few sensilla on the ovipositor that responded to Z-3-hexenyl butyrate, the most effective ligand of HassOR31/HassORco, which explains the low expression of *HassORco* in the ovipositor. However, it does not explain the presence of high expression levels of redundant *HassOR31*. We speculate that in addition to cooperating with HassORco and functioning in olfactory sensation, HassOR31 might function in mediating cell responses to endogenous signaling molecules, regulating neural development, or play other non-chemosensory roles. The similar situation is also found in testes of *A. gambiae*, where some AgORs are abundantly expressed, but AgORco transcript is present at a very low level. It is proved that the AgORs and AgORco are localized to the flagella of *A. gambiae* spermatozoa where Orco-specific agonists, antagonists, and other odorant ligands robustly activate flagella beating in an Orco-dependent process (**Pitts et al., 2014**). Studies on mammals have found that many ORs are present in tissues outside the olfactory system and have diverse functions in several physiological contexts beyond odor recognition (**Wu et al., 2017**). For example, a human testicular OR, hOR17-4 was found to control cellular motility and chemotaxis in sperm cells (**Spehr et al., 2003**).

In addition to the ORs, we also found some GRs and IRs present in the ovipositors of *H. assulta*. The expression of HassGR9, the ortholog of fructose receptor HarmGR4 (**Jiang et al., 2015**), suggests that sugar taste sensilla are also present in the ovipositor of *H. assulta*. Sugar taste sensilla have also been found on the ovipositor of cotton leaf worm *Spodoptera littoralis* (**Seada et al., 2016**). The co-receptors of IR families (HassIR8a, HassIR25a) were also expressed, indicating that IRs may also play chemosensory roles in the ovipositor. There is no evidence in the literature of any ionotropic glutamate receptor other than antennal IRs involved in the chemical sensation of insects.

## The features of chemosensory sensilla on the ovipositor

Previous morphology and electrophysiology studies shown that there were various types of sensilla including mechanical, contact chemosensory, and putative olfactory sensilla distributed on the ovipositor of lepidopteran insects (*Faucheux, 1991*; *Klijnstra and Roessingh, 1986*; *Maher and Thiery, 2004*; *Seada et al., 2016*; *Waladde, 1983*; *Yamaoka et al., 1971*). On *H. assulta* ovipositors we identified two types of chemosensory sensilla with shapes similar to the trichoid and basiconic sensilla on the antenna, respectively. However, unlike traditional olfactory sensilla, these multiporous sensilla also have a large terminal pore, which is characteristic of contact chemosensory sensilla. Therefore, we infer that these two types of sensilla may perform both olfactory and taste functions. The electrophysiological results indicated that some sensilla on the ovipositor responded to the tested odorants though their firing frequency was lower than that on the antennae. A 100-microgram equivalent of the tested compound stimulated around 20 spikes per second. The putative olfactory sensilla identified in the ovipositor of *M. sexta* have similar characteristics (*Klinner et al., 2016*).

In summary, this was the first study to functionally characterize an OR, HassOR31, expressed in *H. assulta* ovipositors. We also characterized sensilla related to the ovipositor OR with a supplemental function to antennal olfactory sensilla. These sensilla were involved in detecting GLV and terpenoid compounds in host plants, and most likely played a role in oviposition site selection of this oligophagous herbivore. Future researches should focus on knocking out *HassOR31* or destroying the related sensilla in the ovipositor and then measuring how insect behaviors are affected. The CRISPR-Cas9 genome editing method provides a good opportunity for us to knock out *HassOR31* to verify its function. Moreover, the pathway of OSN-expressed HassOR31/HassORco projecting into the terminal abdominal ganglion and the function of superfluous HassOR31 in the ovipositor also require further investigation.

## Materials and methods

**Key resources table**

| Reagent type (species) or resource | Designation | Source or reference | Identifiers | Additional information |
|---|---|---|---|---|
| Gene (*Helicoverpa assulta*) | *18S* ribosomal RNA gene | NCBI | GenBank: EU057177.1 | |
| Commercial assay, kit | RNeasy Plus Universal Mini Kit | Qiagen | Cat# 73404 | |
| Commercial assay, kit | Dynabeads mRNA purification kit | Invitrogen | Cat# 61006 | |
| Commercial assay, kit | Q5 High-Fidelity DNA Polymerase | NEB | Cat# M0491 | |
| Commercial assay, kit | M-MLV reverse transcriptase | Promega | Cat# M1701 | |
| Commercial assay, kit | SYBR Premix Ex TaqII | Takara | Cat# RR820 | |
| Commercial assay, kit | T7/SP6 RNA transcription system | Roche | Cat# 10999644001 | |
| Commercial assay, kit | mMESSAGE mMACHINE SP6 | Ambion | Cat# AM1340 | |
| Software, algorithm | Trimmomatic | Trimmomatic | RRID:SCR_011848 | |
| Software, algorithm | FastQC | FastQC | RRID:SCR_014583 | |

*Continued on next page*

*Continued*

| Reagent type (species) or resource | Designation | Source or reference | Identifiers | Additional information |
|---|---|---|---|---|
| Software, algorithm | Trinity | Trinity | RRID:SCR_013048 | |
| Software, algorithm | RSEM | RSEM | RRID:SCR_013027 | |
| Software, algorithm | GraphPad Prism | GraphPad Prism | RRID:SCR_002798 | 7.0 |
| Software, algorithm | ZEN Digital Imaging for Light Microscopy | ZEN Digital Imaging for Light Microscopy | RRID:SCR_013672 | 2012 |
| Software, algorithm | Adobe Illustrator | Adobe systems | RRID:SCR_014198 | CS6 |
| Software, algorithm | pCLAMP software | pCLAMP software | RRID:SCR_011323 | |

## Animal rearing

*H. assulta* were originally collected as larvae in tobacco fields in Zhengzhou, Henan Province, China, and successive generations were maintained in the laboratory under a 16 L: 8 D photoperiod cycle at 26 ± 1℃ and 55–65% relative humidity. The larvae were reared on an artificial diet mainly constituted of wheat germ, yeast, and chili. Pupae were sexed, and males and females were put into separate cages for eclosion. After emergence, moths were fed with 10% honey in water. Two- to three-day old virgin females were used in the experiments.

*Xenopus laevis* frogs were kindly provided by Prof. Qinghua Tao's laboratory in School of Life Sciences, Tsinghua University, Beijing, China, and reared with pig livers as food in our laboratory at 20 ± 1℃. *Xenopus laevis* were anesthetized by submersion in frozen water, and the oocytes were surgically collected before the related experiments. All procedures were approved by the Animal Care and Use Committee of the Institute of Zoology, Chinese Academy of Sciences for the care and use of laboratory animals.

## Transcriptome sequencing and gene identification

One hundred PG-OVs were dissected from virgin female *H. assulta* during the 5th-8th hour of the scotophase and stored in −80℃ freezer until RNA extraction. Total RNA was isolated using RNeasy Plus Universal Mini Kit (Qiagen, Hilden, Germany), in which genomic DNA was removed by gDNA Eliminator. RNA concentration was determined using an ND-2000 spectrophotometer (Nanodrop, Wilmington, DE, USA). RNA integrity was verified on Agilent 2100 BioAnalyzer (Agilent, USA), and mRNA was isolated by magnetic beads with Oligo (dT) from ten μg of total RNA using Dynabeads mRNA purification kit (Invitrogen, USA). In the next step, paired-end RNA-seq libraries were prepared by following Illumina's library construction protocol. The libraries were sequenced on Illumina HiSeq2000 platform (Illumina, USA) in the Beijing Institutes of Biological Sciences, Chinese Academy of Sciences. FASTQ files of raw-reads were produced and sorted by barcodes for further analysis.

Prior to assembly, 2 × 100 bp paired-end raw reads from each cDNA library were processed to remove adaptors, low quality sequences (Q < 20), and reads contaminated with microbes using Trimmomatic package (*Bolger et al., 2014*). The FastQC package was used to verify the quality of resulting trimmed and filtered reads. The clean reads were de novo assembled to produce contigs using *Grabherr et al., 2011* (https://github.com/trinityrnaseq/trinityrnaseq/), the short reads assembling program using default parameters.

We downloaded amino acid sequences of ORs of *H. armigera* from NCBI (https://www.ncbi.nlm.nih.gov/) to construct local database. We then used BlastX with an E-value cut-off of 1e-5 to search the database to identify putative OR transcripts from the transcriptome we sequenced. To evaluate transcript expression abundances, the RSEM (*Li et al., 2017a*; http://deweylab.github.io/RSEM/) package was applied for calculation of the normalized gene expression value FPKM and TPM.

We also sequenced transcriptomes of *H. assulta's* male antennae, female antennae, proboscis, and forelegs. The clean reads from the above libraries were assembled with *H. assulta's* pheromone gland-ovipositor transcriptome as described in *Li et al., 2017b*. FPKM and TPM values of all candidate genes from the different tissues were calculated to indicate the tissue abundance distribution of identified genes.

## Sequence verification of identified genes from transcriptome sequencing

PCR sequencing was used to verify authentication of the identified genes. PCR experiments were conducted in a 25 μL reaction system with Q5 High-Fidelity DNA Polymerase (NEB) by using a thermal cycler. The thermal cycling conditions were set as follows: 98℃ for 30 s; 30 cycles of 98℃ for 10 s, 50℃ for 30 s, and 72℃ for 90 s; and 72℃ for 2 min. PCR products were analyzed on 1.2% agarose gels and then verified by DNA sequencing. Primers for expression analysis were designed according to the sequencing results.

## Expression analyses

To illustrate and compare the expression of HassOR31 in different tissues and organs, semi-quantitative reverse transcription PCR (RT-PCR) and real-time quantitative PCR (qRT-PCR) were conducted. Male antennae, female antennae, female head without antennae, female thorax, female abdomen, female legs, female wings, and pheromone gland–ovipositor were separately collected from 3 to 100 individuals, depending on the size of the organ, and then stored at −80℃. Total RNA was isolated as described above. cDNA was synthesized with M-MLV reverse transcriptase (Promega, Madison, WI, USA) from the total RNA.

The synthesized cDNA was used as a template in RT-PCR reactions with gene-specific primers. PCR was performed under the following conditions: 94℃ for 3 min, 30 cycles of 94℃ for 30 s, 57℃ for 30 s, 72℃ for 30 s, and 72℃ for 5 min. PCR amplification products were run on a 1.2% agarose gel and verified by DNA sequencing. An actin gene fragment was used as the reference to adjust the initial amount of cDNA used in the PCR procedure.

qRT-PCR was conducted using Mx3005P qPCR System (Agilent Technologies, CA, USA). All reactions were performed in triplicate in a total volume of 20 μL containing 10 μL SYBR Premix Ex TaqII (TaKaRa, Otsu, Japan) and 0.4 mM of each primer under the following conditions: 95℃ for 30 s followed by 40 cycles of 95℃ for 5 s, 60℃ for 34 s, and 72℃ for 30 s, 1 cycle 95℃ for 15 s, 60℃ for 1 min, 95℃ for 15 s. Expression levels of all detected genes were calculated using the $2^{-\Delta Ct}$ method, with 18S gene transcript as an internal control for sample normalization. All experiments were repeated three times using three independent RNA samples. The primer sequences are listed in *Supplementary file 2*.

## In situ hybridization

Two-color double in situ hybridizations were performed following protocols reported previously (*Ning et al., 2016*). Primers were designed to synthesize the gene-specific probe sequences from an open reading frame (*Supplementary file 2*). The product sizes of *HassOR31* and *HassORco* were 846 and 834 nucleotides, respectively. Both digoxin (Dig)-labeled *HassOR31* probe and biotin (Bio)-labeled *HassORco* probe were synthesized with DIG RNA labeling kit version 12 (SP6/T7) (Roche, Mannheim, Germany), with Dig-NTP or Bio-NTP (Roche, Mannheim, Germany) labeling mixture, respectively. Antisense and sense probes were generated from linearized recombinant pGEM-T vector using the T7/SP6 RNA transcription system (Roche, Basel, Switzerland) following recommended protocols. RNA probes were subsequently fragmented to an average length of about 300 bp by incubation in carbonate buffer.

Ovipositors were dissected from three-day-old female moths, embedded in JUNG tissue freezing medium (Leica, Nussloch, Germany) and frozen at −80℃ until use. Sections (12 μm) of ovipositors were then mounted on SuperFrost Plus slides (Boster, Wuhan, China). After a series of fixing and washing procedures, 100 μL hybridization solution (Boster, Wuhan, China) containing both Dig and Bio probes was placed onto the tissue sections. After adding a coverslip, slides were incubated in a humid box at 55℃ overnight. After hybridization, slides were washed twice for 30 min in 0.1 × saline sodium citrate (SSC) at 60℃, treated with 1% blocking reagent (Roche) in TBST for 30 min at room

temperature, and then incubated for 60 min with anti-digoxigenin (Roche, Mannheim, Germany) and Streptavidin-HRP (PerkinElmer, Boston, USA). Visualization of hybridization signals was performed by incubating the sections first for 30 min with HNPP/Fast Red (Roche, Mannheim, Germany), followed by three 5 min washes in TBS, with 0.05% Tween-20 (Tianma, Beijing, China) at room temperature with agitation. Then sections were incubated with Biotinyl Tyramide Working Solution for 8 min at room temperature followed by the TSA kit protocols (PerkinElmer, USA). Sections were then washed three more times for 5 min each in TBS with 0.05% Tween-20 at room temperature with agitation. Finally, sections were mounted in Antifade Mounting Medium (Beyotime, Beijing, China). Images were taken using a Carl Zeiss LSM710 confocal microscope (Zeiss, Oberkochen, Germany) and processed using ZEN 2012 software. Adobe Illustrator (Adobe systems, San Jose, CA) was used to arrange figures and the images were only altered to adjust the brightness or contrast.

## Functional analysis of HassOR31

We expressed full length coding sequences of *HassOR31/HassORco*, *HassOR31*, *HassOR31/Hassi-GluR7* in *X. laevis* oocytes and analyzed the oocytes using two-electrode voltage clamping, as previously described (*Jiang et al., 2014*). Oocytes were challenged with 51 chemicals with a concentration of $10^{-4}$ M, including host plant volatile compounds and two principal sex pheromone components (listed in *Supplementary file 3*). Two-electrode voltage clamping was used to detect the whole cell current.

Total RNA and cDNA of ovipositors were obtained as described above. To obtain the full-length coding sequences, PCR was carried out using gene-specific primers with Kozak consensus sequence and Restriction Enzyme cutting site based on the mRNA sequences of *HassOR31*, *HassORco*, and *HassiGluR7*. The primer sequences are provided in *Supplementary file 2*. The PCR program included initial denaturation at 94℃ for 2 min, followed by 35 cycles of 30 s at 94℃, 30 s at 55℃, 90 s at 72℃; and a final extension step of 8 min at 72℃. Then, the coding sequences of *HassOR31*, *HassORco*, and *HassiGluR7* were cloned into pGEM-T easy vector (Promega, Madison, WI, USA), then subcloned into pCS2+ vector. The pCS2+ vectors were linearized by using NotI (Takara Shuzo, Shiga, Japan), cRNAs were synthesized from the linearized pCS2+ vectors with mMESSAGE mMA-CHINE SP6 (Ambion, Austin, TX, USA). The cRNAs were dissolved in RNase-free water and stored at −80℃.

Mature, healthy oocytes were treated with 2 mg/mL of collagenase type I (Sigma-Aldrich) in $Ca^{2+}$-free saline solution (82.5 mM NaCl, 2 mM KCl, 1 mM $MgCl_2$, and 5 mM HEPES, pH 7.5) for 1–2 hr at room temperature. Each oocyte was microinjected with 23.6 nL (50 ng) of the mixture of *HassOR31* and *HassORco* (or *HassiGluR7*) cRNA at a ratio of 1:1 or *HassOR31* cRNA alone. Oocytes injected with RNAase-free water was used as a negative control. Injected oocytes were incubated for 3–4 days at 17℃ in the bath solution (96 mM NaCl, 2 mM KCl, 1 mM $MgCl_2$, 1.8 mM $CaCl_2$, and 5 mM HEPES, pH 7.5) supplemented with 100 μg/mL gentamycin and 550 μg/mL sodium pyruvate.

Whole-cell currents were recorded with a two-electrode voltage clamp. Intracellular glass electrodes were filled with 3 M KCl and presented resistances of 0.2–2.0 MΩ. Signals were amplified with an OC-725C amplifier (Warner Instruments, Hamden, CT, USA) at a holding potential of −80 mV, low-pass filtered at 50 Hz and digitized at 1 kHz. Sodium bicarbonate ($NaHCO_3$) was diluted in Ringer's solution before being introduced to the oocyte recording chamber using a perfusion system. Data acquisition and analysis were carried out with Digidata 1322A and pCLAMP software (Axon Instruments Inc, Foster City, CA, USA). Dose-response data were analyzed using GraphPad Prism 7.

## Scanning electron micrographs

The ovipositors of *H. assulta* were carefully dissected and placed into phosphate buffer (PBS, 0.1 M, pH = 7.2) containing 2.5% glutaraldehyde and fixed at 4℃ for 4 hr. They were then flushed in PBS buffer three times for 10 min each time. The ovipositors were placed in increasing gradient ethanol solutions for dehydration. The ethanol concentrations were 10%, 30%, 50%, 70%, 90%, and 100%, respectively. For each concentration, the ovipositors were rinsed for 30 min. Then, they were ultrasonically cleaned in 100% ethanol for 20 s. Then the ovipositors were treated in isoamyl acetate for 30 min. All the samples were dried using a $CO_2$ critical point dryer (model HCP-2, Hitachi, Tokyo, Japan). The dried samples were stuck to the sample stage in different orientations. The samples

were coated with gold using a Hitachi Sputter Ionizer (model S-4800, Hitachi) for 60–90 s. Finally, the samples were photographed using a Hitachi S-4800 scanning electron microscope.

## Single sensillum recording

A single moth was placed into a 1 mL Eppendorf pipette tip with the narrow end cut off. The moth was gently pushed until its abdomen protruded from the cut end. The ovipositor was extended by gently pressing onto the abdomen, then fixed with dental wax and wrapped tightly with Parafilm. The reference electrode was inserted into the abdomen of the insect, and the sharpened tungsten recording electrode was inserted into the base of the sensilla housed in the ovipositor (*Video 2*). The recorded signals were then amplified through an IDAC interface amplifier (IDAC-4, Syntech, Germany). The software Autospike (Syntech, Germany) was used to store and analyze data.

A continuous stream of purified and humidified air was directed onto the ovipositor (12.5 mL/s) from the outlet of a steel tube (i.d. 6 mm, length 15 cm), positioned 1 cm from the ovipositor. Test odorants were injected into the air stream using a stimulus flow controller (CS-55, Syntech, Germany), which generated 200 ms air pulses through the odor cartridge at a flow rate of 10 mL/s, while a compensating air flow was provided to keep a constant current. The odorants supplied during single sensillum recordings are listed in *Supplementary file 3*. All odorants were diluted to a final concentration of 10 µg/µL (1% w/v) in mineral oil. Ten µL of the diluted odors were pipetted onto a small piece of filter paper (2.5 cm ×0.7 cm) and placed inside a glass Pasteur pipette.

## Oviposition choice test

Newly emerged female and male moths were mixed at a sex ratio of 1:1.3 in 26 cm ×26 cm × 26 cm cubic cages covered with gauze for 3 days to ensure the female moths were fully mated, and then the mated female moths were randomly separated into two groups. One group of insects had their antennae removed just before the experiments, and another group of insects were kept as intact females. Two choice tests were performed.

Choice test 1: Ten females from each group were put into a cylinder cage (diameter 24 cm, height 26 cm). Only the top side of the cage was covered with gauze for females to lay eggs, and all the other sides were covered with black cloth. The gauze side was equally divided into four areas. From the beginning of the scotophase, two fresh hot pepper fruit discs of 1.5 cm diameter were positioned above each of two opposite areas of the gauze, and no pepper discs were put above the other two areas. The pepper discs were supported by a stainless net shelf to avoid the moths inside the cage directly contacting the pepper (*Figure 6—figure supplement 1A*). After 24 hr the number of eggs on each section of gauze was counted. After counting, the gauze was replaced by a new one. The number of eggs laid on each part of the gauze were counted every day for 4 days and the mean number of eggs was calculated. Seven replications were run.

Choice test 2: Oviposition preference of female *H. assulta* to Z-3-hexenyl butyrate was performed in screened cages (1m × 1 m×1 m) as described in *Wu et al., 2019*, fifteen females

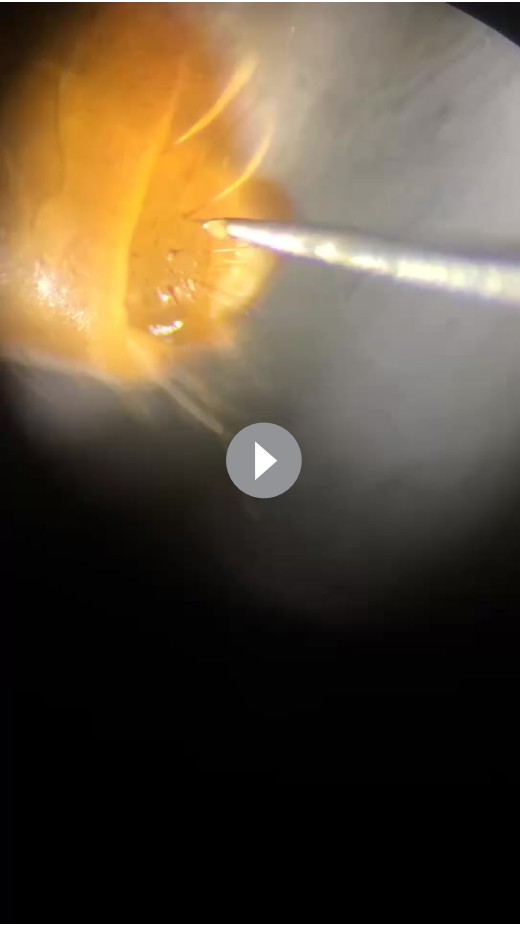

**Video 2.** An exemplary SSR experiment video.
https://elifesciences.org/articles/53706#video2

from each group were put into each cage. In each cage, four fake green plants were respectively placed at the four corners. Two fake plants treated with Z-3-hexenyl butyrate were put in one diagonal, and the other two fake plants with paraffin oil were put in the other diagonal and used as control. Z-3-Hexenyl butyrate were dissolved in paraffin oil and then dropped into a rubber head, which was placed on the leaf of the fake plant (*Figure 6—figure supplement 1B*). The dosage of Z-3-hexenyl butyrate used in the experiment was 100 µg. The number of eggs laid on each fake leaf in 48 hr were counted. The oviposition preference index was calculated as (T−C)/(T+C). T is the number of eggs on the leaves with Z-3-hexenyl butyrate, and C is the number of eggs on the leaves with paraffin oil. Five replications were run.

### Data analysis

Electrophysiological response values (currents and spikes per second) are indicated as mean ± SEM. One-way ANOVA and Turkey's multiple comparisons test were used to compare the responses of HassOR31/HassORco to different tested compounds and single sensillum recording data. Paired t tests were performed to analyze numbers of eggs in oviposition choice tests. When comparing oviposition preference indexes, we used unpaired t tests and differences were considered significant when $p < 0.05$. All data were analyzed using the GraphPad Prism version 7.00 for Mac OS X, GraphPad Software, La Jolla California USA, www.graphpad.com.

## Acknowledgements

We thank our colleagues Zhen Zou, Chao Ning for their assistance in the transcriptome library construction and bioinformatics analysis. We are grateful to Ke Yang and Han Wu, who provide the practical advices on in situ hybridization and behavioral bioassays, respectively. We also thank Meng Xu and Da-Feng Chen for their helps in single sensillum recording. This work was supported by the National Natural Science Foundation of China (Grant No. 31830088 and 31772528), and National Key R and D Program of China (Grant number: 2017YFD0200400).

## Additional information

### Funding

| Funder | Grant reference number | Author |
| --- | --- | --- |
| National Natural Science Foundation of China | 31830088 | Chen-Zhu Wang |
| National Natural Science Foundation of China | 31772528 | Chen-Zhu Wang |
| National Key R and D Program of China | 2017YFD0200400 | Chen-Zhu Wang |

The funders had no role in study design, data collection and interpretation, or the decision to submit the work for publication.

### Author contributions

Rui-Ting Li, Conceptualization, Data curation, Software, Formal analysis, Validation, Investigation, Visualization, Methodology, Writing - original draft; Ling-Qiao Huang, Resources, Data curation, Formal analysis, Investigation, Visualization, Methodology, Writing - original draft; Jun-Feng Dong, Resources, Data curation, Formal analysis, Validation, Investigation, Visualization, Methodology; Chen-Zhu Wang, Conceptualization, Resources, Data curation, Supervision, Funding acquisition, Validation, Methodology, Project administration, Writing - review and editing

### Author ORCIDs

Rui-Ting Li https://orcid.org/0000-0002-4720-6255
Chen-Zhu Wang https://orcid.org/0000-0003-0418-8621

### Ethics

Animal experimentation: All procedures in this study were approved by the Animal Care and Use Committee of the Institute of Zoology, Chinese Academy of Sciences for the care and use of laboratory animals (protocol number IOZ17090-A). The surgery was performed following the protocols reported by Nakagawa and Touhara (2013). The *Xenopus laevis* was anesthetized by bathed in the mixture of ice and water in 30 min, the wounds were carefully treated to avoid infection. Every effort was made to minimize suffering.

### Decision letter and Author response

Decision letter https://doi.org/10.7554/eLife.53706.sa1
Author response https://doi.org/10.7554/eLife.53706.sa2

## Additional files

### Supplementary files

• Supplementary file 1. Expression values of putative chemosensory receptors in the pheromone gland-ovipositor complex of *H. assulta* (*Hass*) and *H. armigera* (*Harm*). OR, odorant receptor; GR, gustatory receptor; IR, antennal ionotropic receptor; iGluR, ionotropic glutamate receptor.

• Supplementary file 2. Primers used for qRT-PCR, RT-PCR, probe synthesis and full-length cDNA cloning.

• Supplementary file 3. Tested compounds for functional analysis of HassOR31.

• Transparent reporting form

### Data availability

Sequencing data has been uploaded to SRA database under PRJNA592822 and SAMN13440804.

The following datasets were generated:

| Author(s) | Year | Dataset title | Dataset URL | Database and Identifier |
|---|---|---|---|---|
| Rui-Ting Li, Ling-Qiao H, Jun-Feng D, Chen-Zhu W | 2020 | Model organism or animal sample from Helicoverpa assulta | https://www.ncbi.nlm.nih.gov/biosample/SAMN13440804 | NCBI BioSample, SAMN13440804 |
| Rui-Ting Li, Ling-Qiao H, Jun-Feng D, Chen-Zhu W | 2020 | Transcriptome sequencing of pheromone gland- ovipositor of Helicoverpa. assulta (a moth) | https://www.ncbi.nlm.nih.gov/bioproject/PRJNA592822 | NCBI BioProject, PRJNA592822 |

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
