## [Decision Letter]

**Acceptance summary:**

The manuscript provides a detailed and innovative view of how *Helicoverpa assulta* perceives plant volatiles with an odorant receptor on the ovipositor. The pictures on egg distribution are a valuable addition, and the new text nicely clarifies the last few issues raised by the reviewer.

**Decision letter after peer review:**

Thank you for submitting your article "A highly expressed odorant receptor in the ovipositor of a moth is involved in detecting host-plant volatiles" for consideration by *eLife*. Your article has been reviewed by three peer reviewers, one of whom is a member of our Board of Reviewing Editors, and the evaluation has been overseen by Ian Baldwin as the Senior Editor. The following individuals involved in review of your submission have agreed to reveal their identity: Markus Knaden (Reviewer #2); Leslie Voshall (Reviewer #3).

The reviewers have discussed the reviews with one another and the Reviewing Editor has drafted this decision to help you prepare a revised submission.

Your manuscript provides extensive data on an odorant receptor (OR) present in the ovipositor/pheromone gland complex of the moth *Helicoverpa assulta*. The study shows that there is one OR that is higher expressed in the olfactory receptor ovipositor/pheromone gland complex than in the antennae of the insect. Yet, the antennae are the best known organs involved in olfaction by moths and many other insects. This yields an intriguing question, i.e. on the involvement of this OR in oviposition decisions by the moths. By expressing the odorant receptor and its co-receptor in a *Xenopus*oocyte model system, 12 plant odours are identified that yield a response. Finally, by electrophysiological analyses and by investigating oviposition behaviour of the moths with intact antennae or after removal of the antennae you provide evidence supporting the role of the odorant receptor in the ovipositor/pheromone gland complex in odour perception and in behavioural decisions of the animals.

The most important criticism is that the final piece of the puzzle is lacking, i.e. the evidence that knocking out the OR specifically eliminates the effects of the plant volatiles on oviposition. Although such information would be really nice, the dataset as it stands is quite extensive and an important step. Developing CRISPR-Cas9 for this system would be a major step in itself. Yet, it will be important to address the value of such a next step in the current manuscript to provide the full evidence, thus supplementing the current step taken, i.e. antennectomy.

In addition, we ask you to provide additional data on the recordings in Figure 2 to substantiate the replication of these findings.

You will find the detailed comments by the three reviewers and ask you to revise the manuscript in response to these comments.

A revised version of the manuscript will be sent out for additional review.

Reviewer #1:

This manuscript presents data on an odorant receptor that is highly expressed in the ovipositor/pheromone gland and that is involved in the selection of oviposition sites. The OR is responsive to a limited number of plant volatiles, none of which seem to be specific for the host plants of the moth. That an odorant receptor is highly expressed in the ovipositor/pheromone gland is remarkable and the data presented here provide a new view of how female moths use plant information during decisions in oviposition site selection.

The study is well performed, albeit that the behavioural experiments are done with a rather limited number of individuals, but the conclusions are straightforward. The antennectomy treatment is a rather crude treatment. Yet, I wonder why the reverse experiment, i.e. ovipositorectomy has not been carried out as well.

Reviewer #2:

In their study the authors present evidence that female moths of *Helicoverpa assulta* evaluate leaves as oviposition sites with olfactory sensilla situated on the moths' ovipositor. By a huge array of experiments (transcriptome analysis of the ovipositor, in situ hybridization of that receptor that is strongly expressed in the ovipositor, functional analysis of this receptor in frog oocytes, SSR experiments with basiconic-like sensilla on the ovipositor with those odorants the oocytes ectopically expressing the receptor responded to, and finally oviposition assays with moths lacking olfactory input via the antennae) the authors convincingly show that the receptor *HassOR31* expressed in sensilla on the ovipositor is governing oviposition choice in this species. While this is not the first report of olfactory sensilla the expression of olfaction-related genes on the ovipositor, it is to my knowledge the most comprehensive analysis and the first one to show, which olfactory receptor is involved. I have only few rather minor concerns:

1) The authors show that moths without antennae still prefer leaves with the major ligand of *HassOR31* added (i.e. the ligand that becomes detected by the olfactory sensilla on the ovipositor). This suggests that olfaction by the ovipositor is sufficient to govern oviposition behavior. It would have been, however, much nicer, if the authors would have included experiments with gene-edited moths, lacking *HassOR31*. Anyhow, the establishment of CRISPR Cas9 in a new species is very time consuming, and I feel that the manuscript even without this critical test for receptor sufficiency is convincing.

2) The manuscript is easy to read but would profit from some language editing. Sometimes the sentences come in the right sequence but are not well connected to each other (just adding "Therefore, we performed…" or "However,…" would often help to make the flow better).

3) I find it very interesting, that the preference indices of moth lacking antennae differ for test with hot pepper (i.e. a complex cue) but not for tests with cis-3-Hexenyl butyrate, i.e. the ligand for the receptor on the ovipositor. This shows that general oviposition is governed by both antennae and ovipositor, but that only cis-3-Hexenyl butyrate seems to be mainly governed by the ovipositor. This important information is somewhat hidden in the text.

Reviewer #3:

The work identifies a moth odorant receptor and its co-receptor *ORco* in the ovipositor using RNA-seq. While the work is reasonably rigorously carried out (though see concerns below), there is no evidence that *HassOR31* is functionally required in oviposition behavior. There is circumstantial evidence to that effect, but nothing in the data confirms it. The transitive argument that an OR is found in ovipositor, responds to X ligand, and X ligand has an influence on oviposition does not prove that this OR regulates egg-laying. Such proof would require genetic disruption of the receptor by genome-editing. The work is therefore primarily of interest to specialists in the field.

---

## [Author Response]

The most important criticism is that the final piece of the puzzle is lacking, i.e. the evidence that knocking out the OR specifically eliminates the effects of the plant volatiles on oviposition. Although such information would be really nice, the dataset as it stands is quite extensive and an important step. Developing CRISPR-Cas9 for this system would be a major step in itself. Yet, it will be important to address the value of such a next step in the current manuscript to provide the full evidence, thus supplementing the current step taken, i.e. antennectomy.In addition, we ask you to provide additional data on the recordings in Figure 2 to substantiate the replication of these findings.You will find the detailed comments by the three reviewers and ask you to revise the manuscript in response to these comments.A revised version of the manuscript will be sent out for additional review.

Many thanks to the editors and three reviewers for their recognition of this work and valuable comments and suggestions. Odorant receptors associated with the detection of volatile chemical signals/cues are typically located on the antennae or maxillary palps of insects. We integrated the transcriptome analysis, and molecular, electrophysiological, and behavioral techniques and methods to reveal that the ovipositor of the moth species *Helicoverpa assulta* bearing a functional OR complex, *HassOR31*-*ORco*, also plays a role in detecting volatile chemical cues.

On your concerns:

1) We agree that if we can bring to the modern technology of CRISPR-Cas9 genome editing, it would directly and fundamentally illuminate the involvement of *HassOR31* in oviposition decisions by the moths.

In fact, our laboratory has been trying to establish CRISPR-Cas9 in this species and its closely related species *Helicoverpa armigera*. However, up to now it is not successful on *H. assulta*: one reason is that this oligophagous species is not easy to be successively reared in the single-pair crossing condition in the lab. Considering the special topic of this study, even if we could knock out *HassOR31* in the whole body of *Helicoverpa assulta*, it would be still hard to explain the function of *HassOR31* expressed in ovipositor because *HassOR31* in the antennae would also been removed. The ideal mutant comparing with the wild type for this study is the insects with no or less expression of *HassOR31* in the ovipositor but normal expression in the antennae, which is even more difficulttobe achieved now in this non-model species. In such a case we have discussed the value of such a next step, and hopefully can reach this goal in the near future.

2) We have added new recordings in Figure 2, and also provided some additional data in two new supplementary figures (Figure 2—figure supplement 1, Figure 2—figure supplement 2). In these figures we can see that in most cases *HassOR31* is expressed alone.

We will explain these in detail in the terms of responding to three reviewers’ comments as follows.

Reviewer #1:[…] rather limited number of individuals, but the conclusions are straightforward. The antennectomy treatment is a rather crude treatment. Yet, I wonder why the reverse experiment, i.e. ovipositorectomy has not been carried out as well.

Many thanks for your comments and suggestions.

1) Yes, the ligands of *HassOR31* including Z-3-hexenyl-butyrate, myrcene, citral, and Z-3-hexenyl acetate are all common plant volatiles, suggesting that the ovipositors owning such a highly expressed OR ensure *H. assulta* females to lay their eggs on the green plant surface. In contrast, its closely related species *H. armigera*, a generalist, has no such strictness on oviposition behaviors, and the neonate larvae are also much more mobile and robust than those of *H. assulta*.

2) Antennectomy was our unwilling choice in the beginning. In the pre-experiments, besides removing the antennae, we had also tried to use the nail polish to seal the sensilla on the surface of the ovipositor, or to seal the sensilla on the antennae, but both failed. We found that the antennectomized female moths could still lay eggs, so we chose this relatively crude treatment. Ovipositorectomy was not considered because the moths without the ovipositor could not lay any eggs.

3) We set up two oviposition preference experiments, one with the host-plant volatile mixture, another with the most effective ligand Z-3-hexenyl-butyrate. In each experiment we arranged 5-7 biological replicates, and in each replicate, we used 10-15 female moths (with or without antennae). Considering it is not easy rearing insect species in the lab, the individuals we used are already a large number in total although the replications are still limited. When we received these comments, we immediately planned to carry out more replications. Unfortunately, the unexpected new coronavirus spreading in China prevented us from doing anything in this period of time.

Reviewer #2:[…] I have only few rather minor concerns:1) The authors show that moths without antennae still prefer leaves with the major ligand of HassOR31 added (i.e. the ligand that becomes detected by the olfactory sensilla on the ovipositor). This suggests that olfaction by the ovipositor is sufficient to govern oviposition behavior. It would have been, however, much nicer, if the authors would have included experiments with gene-edited moths, lacking HassOR31. Anyhow, the establishment of CRISPR Cas9 in a new species is very time consuming, and I feel that the manuscript even without this critical test for receptor sufficiency is convincing.

Thanks for your comments and understanding. As our responses above to the related comments of the editor, gene editing exploring by CRISPR-Cas9 for this species is ongoing in our laboratory, but we still have a long way to go, hence instantly using this technology for this species is impractical at present. In this study, we take advantages of the unusual high expression of *HassOR31* in ovipositor of *H. assulta* to reveal its hidden secret, we validated the co-expression of *OR31* and *ORco* under the sensilla of ovipositor and identified its ligands. Electrophysiological and behavioral experiments proved the related sensilla play a role in insect oviposition preference. To provide the last piece of evidences, we will do our best to knock out the genes by genome-editing in future, as discussed in the article.

2) The manuscript is easy to read but would profit from some language editing. Sometimes the sentences come in the right sequence but are not well connected to each other (just adding "Therefore, we performed…" or "However,…" would often help to make the flow better).

Thanks so much for your comments and suggestions. We have revised the whole manuscript and made the language more fluent to read (such as in the second paragraph of the Introduction, subsections “Tissue expression pattern of *HassOR31*”, “Localization of *HassOR31* and *HassORco* in the ovipositor” and “Oviposition preference of *H. assulta* female”).

3) I find it very interesting, that the preference indices of moth lacking antennae differ for test with hot pepper (i.e. a complex cue) but not for tests with cis-3-Hexenyl butyrate, i.e. the ligand for the receptor on the ovipositor. This shows that general oviposition is governed by both antennae and ovipositor, but that only cis-3-Hexenyl butyrate seems to be mainly governed by the ovipositor. This important information is somewhat hidden in the text.

Thank you so much for this valuable insight. We have added some sentences on this point in the Discussion section as follows: “The behavioral assay with hot pepper volatiles (a complex cue) showed that antennectomized females still preferred to oviposit on sites treated with hot pepper volatiles, but their oviposition preference index was significantly decreased. […] The expression levels of *HassOR31* and other ORs expressed in the ovipositor and antennae may explain such behavioral responses.”

Reviewer #3:The work identifies a moth odorant receptor and its co-receptor ORco in the ovipositor using RNA-seq. While the work is reasonably rigorously carried out (though see concerns below), there is no evidence that HassOR31 is functionally required in oviposition behavior. There is circumstantial evidence to that effect, but nothing in the data confirms it. The transitive argument that an OR is found in ovipositor, responds to X ligand, and X ligand has an influence on oviposition does not prove that this OR regulates egg-laying. Such proof would require genetic disruption of the receptor by genome-editing. The work is therefore primarily of interest to specialists in the field.

Thank you so much for your critical comments. We largely accept your advice. Nevertheless, the primary finding of this study is the olfactory function of *HassOR31* in the ovipositor of *H. assulta*, rather than requirement of this gene in oviposition behavior although the latter is also what we do best to validate. We give thanks to the solid foundation on molecular mechanisms of insect olfaction built by previous studies, especially discovery of the ligand-gated ion channel formed by OR-ORco complex in *Drosophila*. It is by applying this “principle” we used a series of comprehensive methods including transcriptome analysis, in situ hybridization, *Xenopus* expression system and two-electrode voltage-clamp, SSR, and finally oviposition assays to prove the olfactory function of *HassOR31* in the ovipositor of *H. assulta*. It should be of public interest because generally people believe the olfaction organ is only on the head of insects, here we tell the ovipositor can smell in some moths and provide the first example to show which olfactory receptor is involved in such a process.

*Helicoverpa armigera* and *Helicoverpa assulta* are two ideal models in studying olfaction mechanisms of insects. They are closely related species but have big differences in sex pheromone communication and host-plant selection. The genome information of *Helicoverpa armigera* has been reported, while that of *Helicoverpa assulta* has not yet. We are on our way in trying to turn these two moth species into new model organisms with the modern technology of CRISPR-Cas9 genome editing.

We totally agree that the model organisms like *Drosophila* have provided valuable insights into biology including olfaction. The logic used in genetic dissection is quite simple and effective: inactivate an OR gene and measure the behavioral output, and then one can conclude that the OR plays a key role in host-searching behavior. However, the complexity of the system is far too great to be only encoded by a simple correspondence between single genes and neurons, network interactions and the environment also play a role. Yet apart from the technical feasibility in this case, it seems not easy to clarify the role of the OR expressed in the ovipositor by simply knocking the gene out because is also expressed in the main olfactory organ antennae. The behavioral output could be a combined or complementary effect, and therefore transitively dissecting step by step could be another useful approach.